# Epigenetic Factor MicroRNAs Likely Mediate Vaccine Protection Efficacy against Lymphomas in Response to Tumor Virus Infection in Chickens through Target Gene Involved Signaling Pathways

**DOI:** 10.3390/vetsci11040139

**Published:** 2024-03-22

**Authors:** Lei Zhang, Qingmei Xie, Shuang Chang, Yongxing Ai, Kunzhe Dong, Huanmin Zhang

**Affiliations:** 1U.S. Department of Agriculture, Agricultural Research Service, U.S. National Poultry Research Center, Athens, GA 30605, USA; zhanglei01@caas.cn; 2Institute of Special Wild Economic Animal and Plant Science, Chinese Academy of Agricultural Sciences, Changchun 130112, China; 3College of Animal Science, South China Agricultural University, Guangzhou 510642, China; qmx@scau.edu.cn; 4College of Veterinary Medicine, Shandong Agricultural University, Tai’an 271018, China; changshuang@sdau.edu.cn; 5College of Animal Science, Jilin University, Changchun 130062, China; aiyx@jlu.edu.cn; 6Department of Pharmacology and Toxicology, Augusta University, Augusta, GA 30912, USA; kdong@augusta.edu

**Keywords:** small RNA sequencing, microRNA expression, vaccine protective efficacy, Marek’s disease, bursa of Fabricius, chickens, target genes, pathways

## Abstract

**Simple Summary:**

Epigenetic factor microRNAs may modulate vaccine protective efficacy well, supported by experimental data showing that the expression of some microRNAs drastically differs between well-protected and poorly protected genetic lines of chickens. Marek’s disease vaccination and virus challenge trials were conducted in two genetically divergent inbred lines of chickens, and hundreds of microRNAs were identified. However, a small number of miRNAs were observed with significant differential expression per line per vaccination treatment group. One of the vaccines was HVT, which reportedly protects the two lines of birds with a significant difference. The target genes of the differentially expressed miRNAs in response to HVT were reportedly involved with many Gene Ontology terms and pathways, which suggests there is a high complexity of the genetic/epigenetic mechanism modulating vaccine protective efficacy.

**Abstract:**

Epigenetic factors, including microRNAs (miRNAs), play an important role in affecting gene expression and, therefore, are involved in various biological processes including immunity protection against tumors. Marek’s disease (MD) is a highly contagious disease of chickens caused by the MD virus (MDV). MD has been primarily controlled by vaccinations. MD vaccine efficacy might, in part, be dependent on modulations of a complex set of factors including host epigenetic factors. This study was designed to identify differentially expressed miRNAs in the primary lymphoid organ, bursae of Fabricius, in response to MD vaccination followed by MDV challenge in two genetically divergent inbred lines of White Leghorns. Small RNA sequencing and bioinformatic analyses of the small RNA sequence reads identified hundreds of miRNAs among all the treatment groups. A small portion of the identified miRNAs was differentially expressed within each of the four treatment groups, which were HVT or CVI988/Rispens vaccinated line 6_3_-resistant birds and line 7_2_-susceptible birds. A direct comparison between the resistant line 6_3_ and susceptible line 7_2_ groups vaccinated with HVT followed by MDV challenge identified five differentially expressed miRNAs. Gene Ontology analysis of the target genes of those five miRNAs revealed that those target genes, in addition to various GO terms, are involved in multiple signaling pathways including MAPK, TGF-β, ErbB, and EGFR1 signaling pathways. The general functions of those pathways reportedly play important roles in oncogenesis, anti-cancer immunity, cancer cell migration, and metastatic progression. Therefore, it is highly likely that those miRNAs may, in part, influence vaccine protection through the pathways.

## 1. Introduction

Epigenetics is within the stable of heritable phenotypic changes in organisms induced through modification of gene expression and intervention of gene translation other than alteration of the structure of DNA, the genetic code itself [1,2]. Over the last two decades, clear and definite evidence from numerous studies showed that epigenetics is deeply involved in modulating host immunity and tumorigenesis processes upon infections [3,4,5,6]. The epigenetic factor microRNA (miRNA) is reportedly involved in various bioprocesses, including oncogene functions and tumorigenesis suppression [7]. Host cellular miRNAs, like gga-miR-26a, gga-miR-181a, and gga-miR-130a, for instance, play a role in lymphoma cell proliferation suppression [8,9].

Marek’s disease (MD) is a contagious disease of domestic chickens caused by an avian α-herpesvirus [10,11], commonly known as MD virus (MDV). MD has been controlled primarily by the use of MD vaccines since 1970 [10], yet sporadic MD outbreaks take place worldwide [12,13]. Therefore, the commercial companies are clearly aware that MD remains a threat to the poultry industry, as it continues to impose an annual cost of over USD 2 billion to the industry [14]. Commonly used commercial MD vaccines include Herpesvirus of turkeys (HVT), SB-1, and CVI988/Rispens, in addition to others that have been under active tests and development [15]. MD vaccine efficacy is dependent upon multiple factors including host genetics [16]. Our previous studies showed the protective efficacy of a MD vaccine can differ drastically from one genetic line of birds to the next [17]. Advancement in understanding the underlying genetic and epigenetic factors that modulate vaccine protective efficacy against tumor incidence would greatly improve the strategy for design and development of new and more potent vaccines, and therefore, would empower better control of infectious diseases, like Marek’s disease, in the chicken.

Genetic resistance to MD is reportedly attributable to major histocompatibility complex haplotypes [18,19,20,21,22,23], a set of quantitative trait loci (QTLs) [24,25,26], copy number variation [27,28,29], single nucleotide polymorphisms (SNPs) [30,31,32,33], and transcriptomic variation of coding genes [34,35,36]. In addition, like what has been reported in human cancer studies [37,38,39,40,41], epigenetics is shown to play an important role in augmenting resistance and susceptibility to MD [42,43,44,45,46,47,48].

Epigenetic factors include histone modification, DNA methylation, and non-coding RNAs [49]. Examination of histone modifications and differential chromatin marks in chickens in response to MDV challenge revealed significant differences between inbred lines of chickens in their resistance to MD at both cytolytic and latency phases [44,50,51]. In a whole genome histone modification study, Luo et al. found that trimethylations of histone H3K4me3 and H3K27me3 marks were positively and negatively correlated with expression of protein coding genes, respectively, both in MDV-challenged and MDV-unchallenged chickens [50]. These reports strongly suggest that histone modifications are highly likely to be involved in MD and potentially associated with tumorigenesis [52] in chicken. DNA methylation affects DNA transcription, consequently passing on impacts on health and disease status in addition to other phenotypic characteristics [38,53]. In a separate study, Luo et al. found that the promoter DNA methylation level of the CD4 gene was downregulated in response to MDV infection, which was negatively correlated with CD4 gene expression in an MD-susceptible line of chickens [42]. The promoter region of CD30, a key gene likely associated with tumorigenesis in MD, was hypomethylated in response to MDV infection and in MD lymphoma [46,54]. DNA methylation level alteration in response to MDV infection was also detected between a genetic line of chickens highly susceptible to MD and another line relatively resistant to MD [43].

MicroRNA (miRNA) is an important class of short non-coding RNAs. Mature miRNAs are approximately 21–25 nucleotides in length. miRNA negatively regulates gene expression by binding to the 3′-UTR or 5′-UTR of mRNAs to inhibit translation or by initiating mRNA degradation to prevent translation of target genes [55]. Dysregulated expression of microRNAs has been observed in numerous disease states, particularly in human cancers, neurologic disorders, autoimmune diseases, metabolic diseases, cardiovascular diseases, and stress-induced emotional and suicidal behavior disorders [56,57]. It is reported that gga-miR-21 was observed to have significantly upregulated expression in response to MDV infection, and miR-26a expression was consistently downregulated in MD tumors. gga-miR-21 and miR-26a are reportedly known to facilitate an MDV oncogene, *Meq*, in lymphomagenesis and to suppress MD tumor formation, respectively [9,58].

Two highly inbred lines of chickens, known as line 6_3_ and line 7_2_, were developed at the USDA, Agricultural Research Service facility, Avian Disease and Oncology Laboratory (ADOL, East Lansing, MI, USA). These lines have subsequently been maintained at both the ADOL and the US National Poultry Research Center (Athens, Georgia) facilities. The 6_3_ and 7_2_ lines share a common major histocompatibility complex haplotype (*B**2) but differ in susceptibility to avian leukosis virus infection and in resistance to MD [19,59]. Our recent studies revealed that MDV infection induced differential expression of nineteen and nine miRNAs in the two genetic lines of birds, respectively [60], and the MD vaccine HVT and CVI988/Rispens induced differential expression of four and thirteen exclusive microRNAs in the 6_3_ line birds, and 0 and one miRNA in the 7_2_ line birds, respectively [61]. To advance the fundamental understanding of microRNAs and microRNA expression in association with MDV infection post-vaccination, this study was designed to profile miRNAs and to identify miRNAs with dysregulated expression in the primary lymphoid organ, bursa of Fabricius, in the two genetically divergent inbred lines of chickens in response to a very virulent plus MDV (vv+MDV) challenge post-vaccination. Predicted target genes of the differentially expressed miRNAs were subjected to Gene Ontology (GO) terms analysis to elucidate the likely biological significance of the differentially expressed miRNAs, potentially in association with MD vaccine protective efficacy.

## 2. Materials and Methods

### 2.1. Experimental Animals and a Vaccination-Challenge Trial

Chicks on the day of hatching were sampled from two highly inbred lines of White Leghorns, known as line 6_3_ and line 7_2_, maintained in the Avian Disease and Oncology Laboratory at East Lansing, Michigan, U.S.A. Both lines are *B**2 haplotype homozygous, but the lines of chickens drastically differ in genetic resistance to MD. Line 6_3_ is resistant and line 7_2_ is highly susceptible [59]. The two lines of chickens also differ in response to MD vaccines [17].

The sampled chicks from line 6_3_ and line 7_2_ were randomly divided into two treatment groups per line. One group was inoculated with HVT at a dose of 2000 plaque-forming units (PFU) each on the day of hatching, and the other group was inoculated with CVI988/Rispens at the same dosage on the same day. Both vaccinated groups were challenged with a very virulent plus strain of Marek’s disease virus (vv+MDV), known as 648A, at a dose of 500 PFU each on the fifth day post-hatch, intraperitoneally (IP). In addition, a control group was also included with the same sample size for each of the chicken lines under the same conditions, in conjunction with a simultaneously conducted joint project. The corresponding control group datasets (SRA accession numbers SAMN11674924-SAMN11674929 under the PRJNA543524 BioProject (URL https://www.ncbi.nlm.nih.gov/sra/PRJNA543524; accessed 2 January 2024) were used in this study only for computation of the differential expression of miRNAs in response to HVT and CVI988/Rispens vaccination. All chickens used in this study were housed in a BSL-2 experimental facility during the trial. Feed and water were supplied ad libitum. The chickens were observed daily throughout the entire duration of the experiment. This animal experiment was approved by USDA, Avian Disease and Oncology Laboratory Institutional Animal Care and Use Committee (IACUC). The IACUC guidelines established and approved by the ADOL IACUC (April 2005) and the Guide for the Care and Use of Laboratory Animals by the Institute for Laboratory Animal Research (2011) were closely followed throughout the experiment.

### 2.2. Extraction of Total RNA Samples

Three chickens from each treatment group were randomly euthanized at 26 days post-vaccine inoculation. Bursa samples were individually collected, and immediately placed into RNAlater solution (Qiagen, Valencia, CA, USA). The collected samples were stored at −20℃ until extractions of the total RNA samples. Total RNA samples were extracted with TRIzol reagent (Invitrogen, Carlsbad, CA, USA) following the manufacturer’s instructions.

### 2.3. Small RNA Sequencing

Total RNA samples were quantitatively and qualitatively checked with a NanoDrop 8000 Spectrophotometer (Thermo Fisher Scientific, Waltham, MA, USA) and an Agilent 2100 Bioanalyzer (Agilent Technologies, Santa Clara, CA, USA), respectively. Good-quality RNA samples were chosen to construct standard cDNA libraries using Illumina TruSeq Small RNA Library Preparation kits following the manufacturer’s recommendations. Completed libraries were subjected to routine quality control (QC) checks and quantified using a combination of Qubit dsDNA HS and Agilent Bioanalyzer High Sensitivity DNA assays. The libraries were sequenced on an Illumina HiSeq 4000 sequencer using SBS (Sequencing by Synthesis) reagents. Base calling was accomplished by use of Illumina Real Time Analysis (RTA) v2.7.7, and the output of RTA was demultiplexed and then converted to FastQ-format data with Illumina Bcl2fastq v2.19.1 (Illumina, San Diego, CA, USA). The small RNA sequence datasets analyzed and reported in this study were deposited to SRA under NCBI (URL https://www.ncbi.nlm.nih.gov/sra; accessed 1 February 2024). The sequence datasets (accession numbers SAMN11674924 to SAMN11674929) of the unvaccinated lines 6_3_ and 7_2_ control groups used in the comparison analyses were generated simultaneously for a joint project following exactly the same protocols as described above, and were also deposited to SRA under NCBI. The small RNA sequencing operations, including library preparation and preliminary reads quality control, were performed at the Research Technology Support Facility, Michigan State University (East Lansing, MI, USA).

### 2.4. Data Analyses of Small RNA_Seq Reads

The data files of small RNA_Seq reads of all samples that passed QC were subjected to analysis one at a time with the miRDeep* software v3.8 [62], with the default parameters except the adapter sequence and the chicken genome build index files. The adapter sequence used in the analysis is TGG AAT TCT CGG GTG CCA AGG AAC TCC AGT CAC (Illumina), and the chicken genome build index (build_bwt_idx) files were constructed based on the chromosome information of the galGal 5.0 genome build. In addition, the “knownMiR.gff” file used in miRDeep* analysis of this study was the latest “gga.gff3” file at the miRbase download website (URL http://www.mirbase.org/download/; accessed 21 February 2024), which was constructed in accordance with galGal 5.0 assembly. Target genes of differentially expressed miRNAs were predicted using the built-in target gene prediction function in miRDeep*.

### 2.5. Validation of the Small RNA Sequence Reads Data by Droplet Digital PCR

A small number of random samples from the identified miRNAs were subjected to Droplet Digital PCR (ddPCR) analysis to validate the levels of small RNA sequence reads datasets. Primers were customarily designed for the analysis for each of the sampled miRNAs using the mature miRNA sequences following the procedures described by Balcells et al., 2011 [63]. The cDNA samples used in the ddPCR validation were reverse transcribed from the individual total RNA samples using the iScript™ RT Supermix Kit (Cat No. 170-8841) following the manufacturer’s instruction (Bio-Rad (Tokyo, Japan)). A ddPCR reaction of a 25 μL final volume was initially prepared per miRNA per biological sample containing 2 μL of cDNA, 12.5 μL of EvaGreen Supermix (Cat No. 1864034), 0.5 μL of each forward and reverse primer (200 nM; synthesized by Eurofins Genomics, Huntsville, AL, USA), and 9.5 μL of nuclease-free water. Of which, 20 μL was loaded into one of eight sample channels of a DG8™ cartridge (Cat No. 1864008, Bio-Rad). Each oil well was loaded with 70 μL of droplet-generating oil (Cat No. 1864006, Bio-Rad). The loaded DG8™ cartridges were placed on a QX200™ droplet generator (Bio-Rad) to generate the digital droplets. A volume of 40 μL of the generated droplet emulsion per sample was transferred to a well in a 96-well PCR plate followed by polymerase chain reaction with EvaGreen on a C1000™ Thermal Cycler (Bio-Rad). The cycling conditions were 95 °C for 5 min, followed by 40 cycles of 95 °C for 15 s, 58 °C for 60 s, and a final extension step of 98 °C for 10 min. The droplets post-PCR were read well by well on a QX200™ droplet reader (Bio-Rad). PCR-positive and PCR-negative droplets in each of the wells were counted and analyzed with the QuantaSoft™ Software (Version 1.7, Bio-Rad). The primers designed for ddPCR validation of the selected miRNAs are given in Table 1.

### 2.6. Identification of Differentially Expressed miRNAs and GO Terms Enrichment

The number of reads per microRNA for each biological sample were counted using HTSeq [64]. In each of the pairwise comparisons (between the MD-vaccine-inoculated group and the control group within each chicken line, between the two MD-vaccinated groups within each chicken line, and between the two chicken lines within each vaccinated group), differentially expressed microRNAs were identified by use of a custom R script encompassing the DESeq R package (2.1.0). Filter criteria of *FDR* < 0.05 and FC > 2 were enforced. For some of the differentially expressed miRNAs that ended up with a zero-statistic estimate for a normalized average of transcripts per million (TPM) in a contrast, an arbitrary small value of 5 was assigned to substitute for the zero to compute a numeric fold change, and then Log_2_ fold change values were computed for easier comprehension of the estimates. To better understand the functional involvements of the identified microRNAs differentially expressed in response to the vaccination, predicted target gene lists of differentially expressed microRNAs for each of the contrasts were subjected to GO terms and pathway analysis using the g:Proflier (URL http://biit.cs.ut.ee/gprofiler/index.cgi; accessed 2 February 2024) online tools with the following options: Organism: Gallus gallus; Statistical domain scope: All known genes; Significance threshold: Bonferroni correction; User threshold: 0.01 [65].

## 3. Results

### 3.1. Small RNA Sequence Reads, Reads Quality, and Deposition to NCBI

Small RNA sequencing generated an average of 35.7 million pass-filter (PF) reads (the number of clusters that passed Illumina’s “Chastity filter”) per biological sample. The PF reads ranged from 23 to 50.8 million with Phred Quality Scores above 30 for the 12 bursal samples of the line 6_3_ and 7_2_ chickens vaccinated with either HVT or CVI988/Rispens and followed by vv+MDV (648A) challenge. The numbers of PF reads of each biological sample for both chicken lines and treatment groups are listed in Table 2. The raw sequence datasets of this study are available at the Sequence Read Archive (SRA) under National Centre for Biotechnology Information (PRJNA544273-SRA-NCBI (nih.gov)).

### 3.2. Validation of the Small RNA Sequence Reads Data by ddPCR

A randomly selected small set of miRNAs was subjected to ddPCR analysis using custom-designed primers (Table 1). The absolute quantification of the four randomly selected miRNAs in the same sets of total RNA samples by ddPCR was analyzed against the corresponding normalized PF reads (TPM) of the small RNA sequence reads data with a bivariate model. The correlation coefficients between the two sets of expression data for the examined miRNAs ranged from r = 0.87 to r = 0.98, with a statistic of *p* < 0.001 (Figure 1). This provided highly positive support in validation of the small RNA sequence reads data generated in this study and the estimates of the miRNA expression derived from the small RNA sequence datasets.

### 3.3. MicroRNA Profiles of the Chicken Line by Vaccine Treatment Groups

A total of 630 unique miRNAs were identified in bursal samples of the two lines of chickens among all the treatment groups. Close to one-third of the identified miRNAs were known chicken miRNAs and the rest were novel miRNAs (Appendix A). A total of 481 and 524 known and novel miRNAs, respectively, were identified in samples of the line 6_3_ chickens vaccinated with HVT and CVI988/Rispens followed by MDV challenge. A total of 457 and 510 known and novel miRNAs, respectively, were identified in samples of line 7_2_ chickens subjected to the same vaccination and MDV challenge. The total numbers of known and novel miRNAs identified in the birds of the lines 6_3_ and 7_2_ in response to HVT or CVI988/Rispens vaccination followed by MDV challenge are summarized in Table 3 for each line by treatment group. The detailed information, including miRNA name (miR_ID), hairpin loci, mature loci, precursor sequence (Sequence), mature sequence (Mature.miR), and normalized number of reads (TPM) is given in Appendix A. A heatmap was constructed, which offers a partial scope of visual illustration of expressional differences and similarities of the identified miRNAs observed among the four line-by-vaccine treatment groups: line 6_3_ vaccinated with HVT or CVI988/Rispens followed by MDV challenge, and line 7_2_ vaccinated with HVT or CVI988/Rispens followed by MDV challenge (Figure 2). As shown, this subset of identified miRNAs most differed in expression between the line 6_3_ and line 7_2_ chickens subjected to the CVI988/Rispens vaccination followed by MDV challenge. The HVT-vaccinated and MDV-challenged line 6_3_ and line 7_2_ groups did differ from each other, but both differed more from the CVI988/Rispens-vaccinated line 7_2_ group than the CVI988/Rispens-vaccinated line 6_3_ group followed by MDV challenge.

A Venn diagram depicts the number of miRNAs identified uniquely within each of the four line-by-vaccine treatment groups, or in common between two treatment groups, or among three or all four treatment groups (Figure 3). The identified numbers of unique miRNAs ranged from 10 to 28 among the four treatment groups, with a dominantly large number of 346 miRNAs in common among all four treatment groups. The HVT-vaccinated MDV-challenged line 6_3_ and line 7_2_ groups were observed with 5 miRNAs in common, the smallest number of identified miRNAs between any two groups of the line-by-vaccine treatment groups; the largest number was 34 miRNAs in common, observed both between the line 6_3_ HVT and CVI988/Rispens-vaccinated MDV-challenged groups and between the CVI988/Rispens-vaccinated MDV-challenged line 6_3_ and line 7_2_ groups.

### 3.4. Differentially Expressed miRNAs between Treatment Groups

Both HVT and CVI988/Rispens vaccination followed by the vv+MDV (648A) challenge induced differentially expressed miRNAs in contrast to control groups of the line 6_3_ and line 7_2_ birds. The control groups of birds were hatched on the same day as the vaccine and MDV treatment groups of birds, and were housed and reared under the same conditions; however, neither were vaccinated nor MDV challenged. The majority of the differentially expressed miRNAs were novel, along with a few of the known chicken miRNAs (Table 4).

#### 3.4.1. Differentially Expressed miRNAs in Response to HVT Vaccination Followed by vv+MDV Challenge in the Line 6_3_ and 7_2_ Chickens

HVT vaccination followed by the vv+MDV challenge induced seven and six differentially expressed miRNAs in bursae of the line 6_3_ and 7_2_ chickens, respectively, 21 days post-MDV inoculation. Five out of the seven differentially expressed miRNAs in line 6_3_ birds in response to HVT vaccination and MDV challenge were significantly upregulated (*p* < 6.5 × 10^−5^), with a range of Log_2_ fold change (FC) from 3.04 to 12.76. The other two miRNAs were significantly downregulated (*p* < 3.1 × 10^−5^), with Log_2_ FC of −6.96 and −8.07, respectively. All the differentially expressed miRNAs in the line 7_2_ birds were significantly upregulated (*p* < 2.5 × 10^−4^) except one, a novel miRNA (novelMiR_692) with a Log_2_ FC of −11.61. The upregulated miRNAs in the line 7_2_ birds were observed with Log_2_ FC ranging from 5.87 to 12.47 (Table 4).

#### 3.4.2. Differentially Expressed miRNAs in Response to CVI988/Rispens Vaccination Followed by vv+MDV Challenge in the Line 6_3_ and 7_2_ Chickens

CVI988/Rispens vaccination followed by vv+MDV challenge induced fourteen and nine differentially expressed miRNAs in bursae of the line 6_3_ and 7_2_ chickens, respectively, 21 days post-MDV inoculation. Thirteen out of the fourteen differentially expressed miRNAs in the line 6_3_ birds were significantly upregulated (*p* < 8.8 × 10^−4^), with Log_2_ FC ranging from 1.49 to 13.94. The other miRNA (novelMiR_91) was significantly downregulated (*p* < 7.6 × 10^−5^), with a Log_2_ FC of −8.0. In the bursae of the line 7_2_ birds, four out of the nine differentially expressed miRNAs were significantly upregulated (*p* < 1.9 × 10^−14^). The Log_2_ FCs ranged from 2.89 to 13.64. The other five differentially expressed miRNAs were significantly downregulated (*p* < 6.1 × 10^−4^), with Log_2_ FCs ranging from −1.33 to −11.98 (Table 4).

#### 3.4.3. Differentially Expressed miRNAs between CVI988/Rispens and HVT Vaccination Groups Followed by vv+MDV Challenge within Each Line of Chickens

By comparison between the treatment groups of CVI988/Rispens and HVT vaccinations followed by MDV challenge, only a few miRNAs were identified within each of the lines of birds that were differentially expressed. The novelMiR_508_1, novelMiR_508_2, and novelMiR_508_3 (identical in their mature miRNA sequence but different at the hairpin loci and mature loci; see Appendix A for details) were significantly upregulated in expression (*p* < 1.2 × 10^−4^) in response to CVI988/Rispens vaccination and MDV challenge with a Log_2_ FC of 6.32, in contrast to HVT vaccination followed by MDV challenge in the line 6_3_ birds. By the same comparison, three miRNAs (gga-mir-30a*, gga-mir-205b, and novelMiR_215) were significantly downregulated in expression (*p* < 2.5 × 10^−6^) in line 7_2_ birds. The Log_2_ FC ranged from −6.15 to −11.10 (Table 5).

#### 3.4.4. Differentially Expressed miRNAs between Line 6_3_ and Line 7_2_ Chickens within HVT or CVI988/Rispens Vaccination Followed by vv+MDV Challenge

Five and twenty-two differentially expressed miRNAs were identified between line 6_3_ and line 7_2_ birds that were vaccinated with HVT and CVI988/Rispens, respectively, followed by MDV challenge.

Two of the five differentially expressed miRNAs (novelMiR_18 and novelMiR_97_2) were expressed significantly higher (*p* < 1.5 × 10^−4^; Log2 FC: 7.23 and 2.46, respectively), and the other three (gga-mir-1684a, novelMiR_215, and novelMiR_1062) were significantly lower (*p* < 1.5 × 10^−4^; Log_2_ FC: −5.85 to −7.97) in line 6_3_ birds than in line 7_2_ birds vaccinated with HVT followed by MDV challenge.

Ten of the twenty-two differentially expressed miRNAs were observed with significantly higher expression (*p* < 1.1 × 10^−3^), and the other 12 miRNAs had significantly lower expression (*p* < 1.7 × 10^−3^), in line 6_3_ birds than the line 7_2_ birds vaccinated with CVI988/Rispens followed by MDV challenge. The 10 miRNAs having significantly higher expression in line 6_3_ birds were observed with Log_2_ FCs ranging from 1.16 to 14.06. The 12 miRNAs having significantly lower expression in line 6_3_ birds were observed with Log_2_ FCs ranging from −1.11 to −8.90 (Table 6).

### 3.5. Gene Ontology (GO) Terms Enrichment of Predicted Target Genes of the Differentially Expressed miRNAs

A total of 2035 and 2134 target genes were predicted for the differentially expressed miRNAs of line 6_3_ and line 7_2_ birds, respectively, in response to the HVT vaccination followed by MDV challenge treatment in contrast to each line’s control group. These target genes were highly enriched in a total of 1729 and 1739 GO terms and pathways, respectively. The target genes were highly enriched in biological process (BP) and molecular function (MF) terms in common between the two HVT-MDV treatment groups of the two chicken lines (6_3_ and 7_2_). The BP terms included the following: (a) cell and cellular processes like cell adhesion, communication, development, differentiation, migration, cell morphogenesis, motility, cell death, cell–cell and Wnt signaling, cell surface receptor signaling pathway, cellular component processes including homeostasis, assembly, biogenesis, morphogenesis, cellular localization and metabolic process, cellular response to stimulus and stress; (b) epithelial processes, like cell differentiation, tube morphogenesis, and development; (c) immune system development; (d) intracellular processes, like protein transport and signal transduction; and (e) negative and positive regulation involving cellular processes, signal transduction, signaling, transcription, apoptotic process, cell communication, gene expression, intracellular signal transduction, and protein phosphorylation. The MF terms included binding processes like ATP binding, DNA bindings, enzyme binding, identical protein binding, kinase and lipid bindings, nucleic acid and nucleotide bindings, protein and protein domain-specific bindings, signaling receptor binding, and small molecule binding. There was a total of 182 GO terms enriched exclusively with target genes of the line 6_3_ HVT-MDV treatment group. Those GO terms included a variety of channel activity terms like ion and calcium channel activity, gated channel activity, and voltage-gated channel activity, and transmembrane transport activity like active transmembrane transporter activity and monovalent inorganic cation transmembrane transporter activity under the MF category. More GO terms under the BP category were observed, which included angiogenesis, regulation of response to external stimulus, Wnt signaling pathway, negative regulation of cell motility and migration, cell–cell adhesion, immune system process, and transmembrane receptor protein serine/threonine kinase signaling pathway (Appendix A). The GO terms for each of the functional categories, which were enriched with target genes of differentially expressed miRNAs of the line 6_3_ and line 7_2_ HVT-vaccinated treatment group and MDV-challenged treatment group of birds, are visually depicted in the top and middle Manhattan-like plots in Figure 4, respectively.

A direct comparison between the line 6_3_ and line 7_2_ HVT-MDV treatment groups resulted in identification of five differentially expressed miRNAs (Table 6), which predictively target a sum of 774 genes. Those 774 target genes were enriched collectively in 907 GO terms, of which 32 out of a total of 45 MF GO terms are highly involved with binding activities ranging from ion and small molecule binding to enzyme and protein binding activities. Close to half of the BP GO terms are involved with regulation processes including regulation of cellular process, transduction by RNA polymerase II, cell communication, signaling, signal transduction, cell mortality, and cell differentiation. Some of the target genes were also enriched in key KEGG pathways, which included MAPK signaling pathway, TGF–beta signaling pathway, ErbB signaling pathway, and EGFR1 signaling pathway. The target genes of these five differentially expressed miRNAs were especially and highly more enriched in the Transfac (TF) GO terms than any other category of GO terms (Figure 4, bottom plot), which included the critical GO terms of the proto-oncogene c-Jun and c-Fos motifs, as well as the cancer-associated Smad3 motifs (Appendix A).

The GO terms enrichment results for target genes of differentially expressed miRNAs for the rest of the treatment groups and comparisons are detailed in Appendix A, which include treatment groups of line 6_3_ and line 7_2_ birds subjected to CVI988/Rispens vaccination followed by MDV challenge (Appendix A and Appendix A, respectively), the comparisons between the line 6_3_ CVI988/Rispens-MDV and HVT-MDV groups (Appendix A), the line 7_2_ CVI988/Rispens-MDV and HVT-MDV groups (Appendix A), and between the line 6_3_ and line 7_2_ groups subjected to CVI988/Rispens-MDV groups (Appendix A).

## 4. Discussion

As one of the avian tumor virus-induced diseases, Marek’s disease has been well under control since the 1970s in most parts of the world, which is primarily attributable to the wide use of MD vaccines in poultry flocks wherever applicable [11]. The commonly used commercial MD vaccines include the gold-standard MD vaccine, CVI988/Rispens; the very first anti-tumor vaccine, HVT; and the MDV-2 vaccine, SB-1 [66,67]. While most, if not all, researchers and industry professionals fully recognize the great good that MD vaccines have done for the poultry industry, few, if any, claim to thoroughly understand the mechanism of how MD vaccines protect against MDV-induced tumorigenesis and tumor formation in chickens [68]. The reality that this paradox persistently remains bars the advancement of knowledge-based new vaccine design and development, as well as better control of MD in poultry.

Protective efficacy of different vaccines varies [69,70,71] and protective efficacy of a single MD vaccine also differs between genetic lines of chickens due to host genetics [16,17]. The highly inbred line 6_3_ and line 7_2_ chickens used in this study, sharing a common major histocompatibility complex haplotype (*B*2*), strikingly differ in their ability to convey protective efficacy by up to 40% in response to CVI988/Rispens and up to 82% in response to HVT vaccination against very virulent plus MDV challenge [17,72]. The observed phenotypic differences in the ability to convey protective efficacy in response to vaccination against MDV-induced tumor formation may result from a rather complex mechanistic system potentially involving host genomic and epigenomic variation and vaccine by chicken line interactions [61,73]. This point is well indicated, in turn, by the fact that the MD vaccine’s immunologic mechanism of protection remains poorly understood [74]. The understanding of genomic and epigenomic mechanisms underlying vaccine protective efficacy is even more limited.

Mounting evidence from research of various biological fields has demonstrated that epigenetic factors like DNA methylations, histone modifications, and non-coding short RNAs, particularly microRNAs, play a pivotal role in differentiation of immune cell subsets and immune functions of those immune cells, which result in alteration of immune responses in reaction to foreign antigens. It is also reported that microRNAs regulate a wide array of cellular processes including, but not limited to, cell proliferation, differentiation, and apoptosis. Dysregulated expression of miRNAs is reportedly associated with malignant transformation and tumor formation [75,76]. To investigate the potential roles that miRNAs play in modulating immune response and, subsequently, tumor incidence post-MD vaccination and MDV challenge, two highly inbred lines of White Leghorns, the lines 6_3_ and 7_2_, were profiled for miRNAs and compared to identify differentially expressed miRNAs in response to HVT or CVI988/Rispens vaccination followed by a vv+MDV challenge. Over six hundred miRNAs were identified among a set of four chicken line by vaccine treatment groups (line 6_3_ HVT+MDV, line 6_3_ CVI988/Rispens+MDV, line 7_2_ HVT+MDV, and line 7_2_ CVI988/Rispens+MDV). Although the total numbers of miRNAs identified within each of the line 6_3_ and line 7_2_ groups subjected to either HVT+MDV or CVI988/Risepens+MDV treatment were similar (Table 3), the numbers of differentially expressed miRNAs in response to CVI988/Rispens+MDV (in contrast to each line’s non-vaccinated non-challenged control group) differed (14 and 9) between the line 6_3_ and line 7_2_ groups (Table 4). It is also noticeable that both HVT+MDV and CVI988/Rispens induced four miRNAs (gga-mir-19b*, gga-mir-425-5p, novelMiR_530, and novelMiR_547) in common with significantly upregulated expression and one miRNA (novelMiR_91) with significantly downregulated expression in common within the line 6_3_ birds (Table 4). Since HVT induced equally good or even better protection of the line 6_3_ birds than the CVI988/Rispens vaccination [72,73], all or some of these five miRNAs dysregulated in expression between the line 6_3_ and line 7_2_ birds in response to the same HVT and MDV treatment would be highly probably associated with the previously observed highly protective efficacy against MD in the line 6_3_ birds [17,72]. Furthermore, of the five differentially expressed miRNAs, the two miRNAs novelMiR_530 and novelMiR_91 were identified in the HVT+MDV treatment group of the line 6_3_ birds but not in the same treatment group of the line 7_2_ birds. Therefore, these two miRNAs might play an even more critical role in modulating the protective efficacy of HVT vaccination in birds like those of line 6_3_ [17,72,73].

Emerging evidence also shows miRNAs play key roles in fine-tuning critical biological processes that affect host immune homeostasis, and consequently infectious diseases and cancer development [77,78,79,80]. It is also shown that tumor viruses are capable of, on one hand, modulating the expression of cellular miRNAs to facilitate their own infection processes by invading the host immune system, and, on the other, to dysregulate oncogenes and tumor suppressor genes [81]. Dysregulated expression of miRNAs in response to virus infection is reportedly associated with cancerous disease development [82,83,84] and host antiviral immunity [85]. It has been shown that the line 6_3_ birds are capable of conveying over 80% protection against vv+MDV-induced MD in response to HVT vaccination, while the line 7_2_ birds are virtually incapable of conveying any protection, or convey a poorly minimal protection, against the same vv+MDV-induced MD in response to HVT [72]. Seven and six differentially expressed miRNAs were identified within the line 6_3_ and 7_2_ birds, respectively, in response to HVT vaccination and vv+MDV challenge in this study (Table 4), which were primarily or totally different from the sets of differentially expressed miRNAs identified in the two lines of birds in response only to HVT vaccination or only to MDV challenge alone (Appendix A). These results may have shown that the induced dysregulation of miRNA expression in response to HVT and MDV treatment is not a simple sum of the HVT treatment plus the MDV treatment.

Direct comparison between the HVT+MDV treatment groups of the line 6_3_ and line 7_2_ birds identified five differentially expressed miRNAs. The same comparison between the CVI988/Rispens+MDV treatment groups of line 6_3_ and line 7_2_ birds identified 22 differentially expressed miRNAs (Table 6). Only two of the differentially expressed miRNAs (gga-miR-1684a and novelMiR-1062) were in common between the two comparison sets of HVT+MDV and CVI988/Rispens+MDV treatment groups.

Gene Ontology analysis of the target genes of the five differentially expressed miRNAs between the line 6_3_ HVT+MDV treatment group and the line 7_2_ HVT+MDV group indicated that those miRNAs, through their target genes, are likely involved with multiple signaling pathways, including the MAPK signaling pathway, which elicits many responses in cells evoked by factors including environmental changes and plays a major role in oncogenesis [86]; the TGF-β signaling pathway, which regulates development, homeostasis, and tissue repairments, and reportedly plays a major role in cancer suppression [87]; the ErbB signaling pathway, which promotes autophosphorylation and subsequent downstream signaling cascades through binding with numerous signal transducers and was confirmed as having an important role involving cancers [88,89,90]; and the EGFR1 signaling pathway, which reportedly plays a role in immune defense against pathogen infection, diverse cellular processes, including cell apoptosis, proliferation, differentiation, migration, and the cell cycle, and immune regulation in both vertebrates and invertebrates [91]. Further research is warranted to clarify the details by providing additional experimental evidence as proof of the mechanism used by the vaccine to achieve the various levels of protection. The findings of this study suggest that these five microRNAs are likely to be instrumental for the host epigenetics to interpolate between vaccine protection and virus-induced disease incidence collectively through varied GO terms and pathways including signaling pathways, with which the microRNAs’ target genes are involved.

## Figures and Tables

**Figure 1 vetsci-11-00139-f001:**
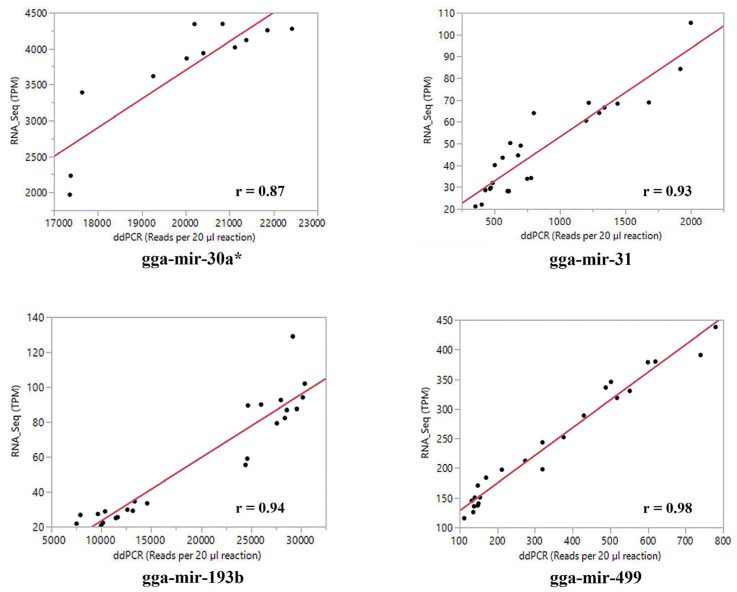
A bivariate plot demonstrating the moderately high association between the droplet digital PCR-generated absolute quantifications of miRNA expressions and the normalized small RNA sequence reads data (TMP) of a set of randomly picked miRNAs out of the identified miRNAs of this study. The result suggests that the small RNA sequence reads datasets generated and analyzed in this study were statistically valid.

**Figure 2 vetsci-11-00139-f002:**
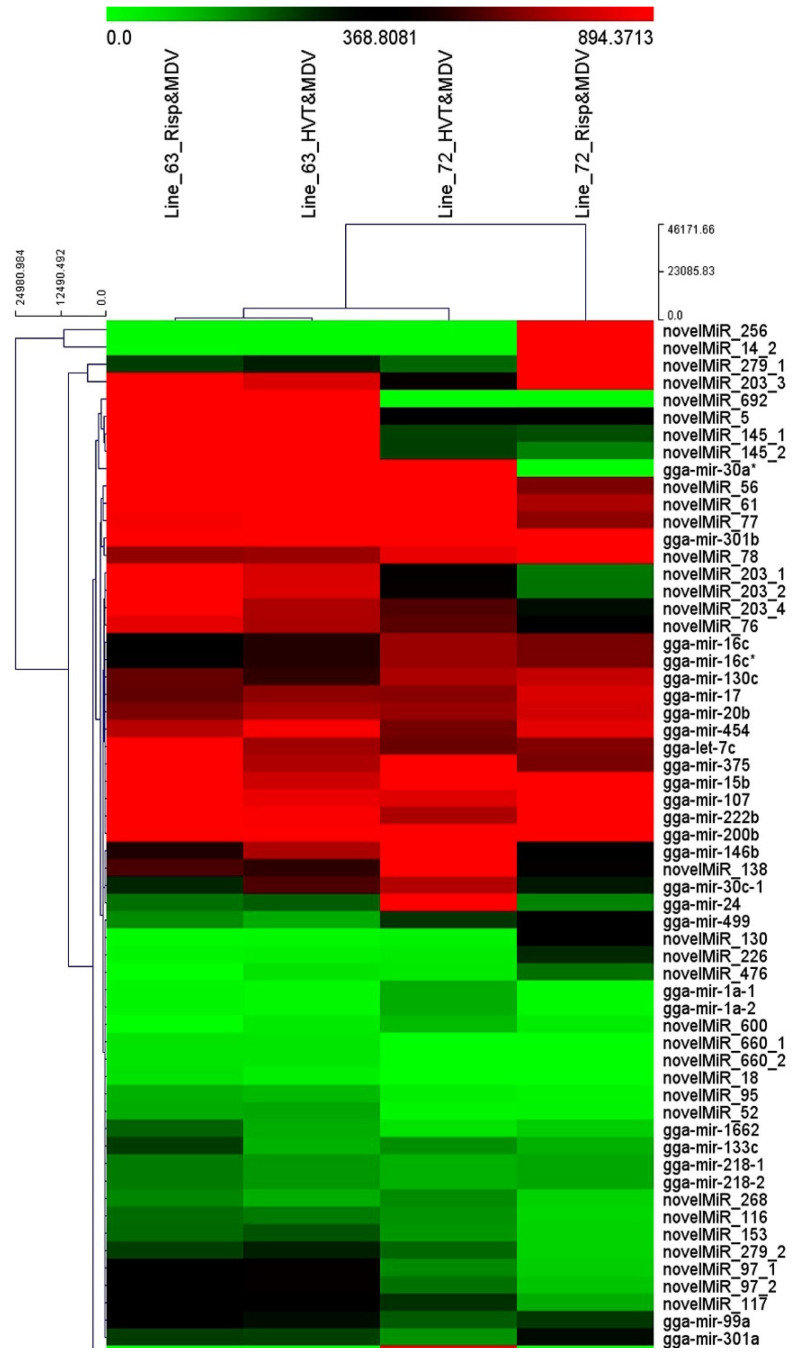
A heatmap graphically demonstrating the identified miRNA expression levels among the four treatment groups. A subsample of identified miRNAs from each of the four line-by-vaccine and MDV treatment groups—line 6_3_ vaccinated with HVT or CVI988/Rispens (Risp), and line 7_2_ vaccinated with HVT or Risp (all followed by Marek’s disease virus challenge)—are plotted to illustrate the expressional differences (normalized counts) and similarities. The expressional levels of the identified miRNAs of the line 6_3_ Risp-vaccinated and MDV-challenged treatment groups were most distant from the Risp-vaccinated and MDV-challenged line 7_2_ group; the line 6_3_ and line 7_2_ HVT-vaccinated MDV-challenged treatment groups did differ from each other, but both were more distant from the Risp-vaccinated MDV-challenged line 7_2_ than the line 6_3_ Risp-vaccinated MDV-challenged treatment groups.

**Figure 3 vetsci-11-00139-f003:**
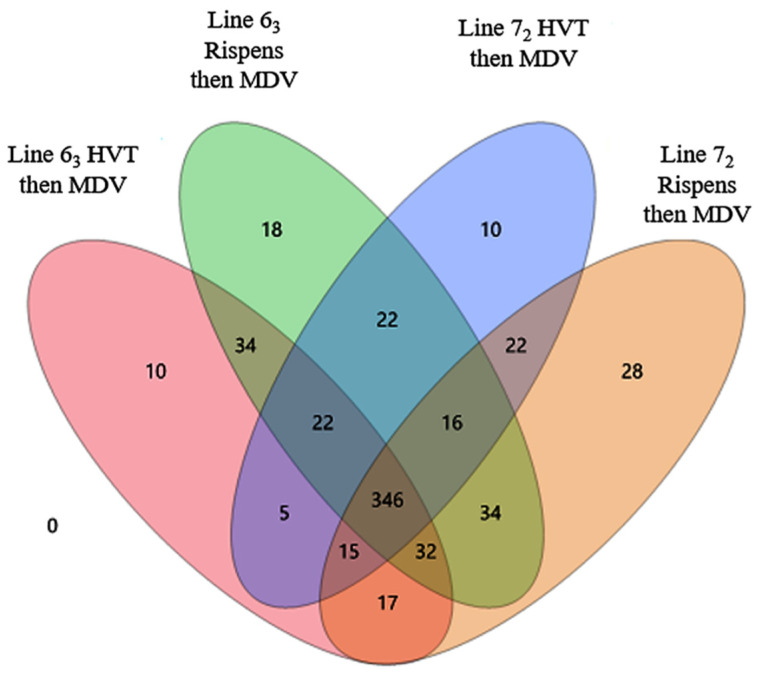
A Venn diagram depicting the numbers of identified miRNAs within each, between two, and among three or all four line-by-vaccine treatment groups. As shown, the numbers of identified miRNAs unique within each of the treatment groups ranged from 10 to 28. More than half (346) of the identified miRNAs (630) were observed in common in all four line-by-vaccine treatment groups.

**Figure 4 vetsci-11-00139-f004:**
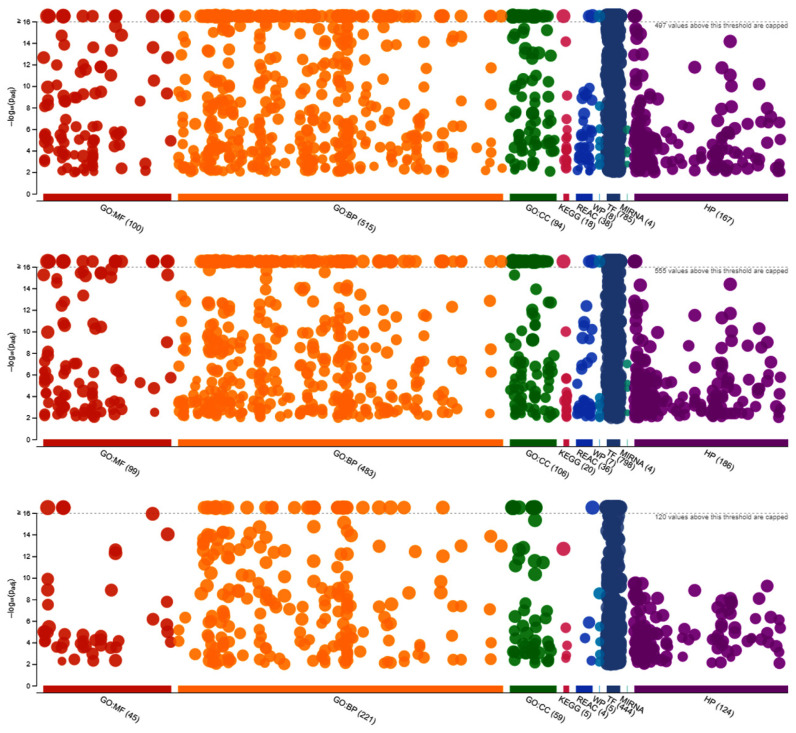
Manhattan-like plots depicting GO terms (MF: molecular function, BP: biological process, CC: cellular component) and pathways (KEGG: Kyoto Encyclopedia of Genes and Genomes, REAC: reactome, WP: WiKiPathways, TF: Transfac, MIRNA: miRTarBase, HP: human protein ontology) of target genes of the differentially expressed miRNAs induced by a combination of HVT Marek’s disease vaccine and Marek’s disease virus (MDV) inoculations: (**top**) line 6_3_ HVT plus MDV treatment group versus line 6_3_ control group; (**middle**) line 7_2_ HVT plus MDV treatment group versus line 7_2_ control group (all the control groups of birds were neither vaccinated nor MDV challenged); (**bottom**) line 6_3_ HVT plus MDV treatment group versus line 7_2_ of the same treatment group. The numbers of GO terms and pathways differed little between the top and the middle plots. The bottom plot, however, showed there were significant numbers of GO terms and pathways involved with the target genes of differentially expressed miRNAs between the line 6_3_ and line 7_2_ birds subjected to the same vaccination and MDV challenge.

**Table 1 vetsci-11-00139-t001:** Customarily designed primers used in ddPCR validation of the RNA sequencing reads data of the randomly picked miRNAs.

miRNA	Sequence	Forward Primer (5′→3′)	Reverse Primer (5′→3′)
gga-mir-30a*	TGTAAACATCCTCGACTGGA	GCGCAGTGTAAACATCCTCGA	CAGGTCCAGTTTTTTTTTTTTTTTCCAGT
gga-mir-31	AGGCAAGATGTTGGCATAGCTG	GCAGAGGCAAGATGTTGGCAT	CAGGTCCAGTTTTTTTTTTTTTTTCAGCTA
ga-mir-193b	AACTGGCCCACAAAGTCCCGCT	CGCAGAACTGGCCCACAAAG	GTCCAGTTTTTTTTTTTTTTTAGCGGGACT
gga-mir-499	TTAAGACTTGTAGTGATGTTT	AGCGCAGTTAAGACTTGTAGTGAT	AGCAGGTCCAGTTTTTTTTTTTTTTTAAACAT

**Table 2 vetsci-11-00139-t002:** Sequenced samples and small RNA sequence reads of the samples.

Chicken Line	Treatment ^1^	Sample No.	Pass-Filter (PF) Reads	PF Reads% ≥ Phred Quality Score 30
Line 6_3_	HVT & MDV	1	49,634,171	96.7%
	HVT & MDV	2	38,184,563	97.1%
	HVT & MDV	3	44,927,209	97.3%
	Risp. & MDV	1	23,024,936	96.2%
	Risp. & MDV	2	26,589,212	96.6%
	Risp. & MDV	3	50,802,611	95.9%
Line 7_2_	HVT & MDV	1	28,507,647	97.3%
	HVT & MDV	2	31,645,519	96.5%
	HVT & MDV	3	37,376,612	96.9%
	Risp. & MDV	1	33,373,070	96.9%
	Risp. & MDV	2	33,949,611	96.7%
	Risp. & MDV	3	30,867,386	96.8%

^1^ Risp.: CVI988/Rispens; HVT: Herpesvirus of turkeys; MDV: Marek’s disease virus.

**Table 3 vetsci-11-00139-t003:** Number of miRNAs identified in two highly inbred lines of White Leghorn layers in response to Marek’s disease vaccination and MDV challenge.

Chicken Line	Treatment ^1^	Number of Identified miRNAs		
		Known	Novel	Sub-Total
Line 6_3_	HVT & MDV	187	294	481
	Risp. & MDV	190	334	524
Line 7_2_	HVT & MDV	182	275	457
	Risp. & MDV	186	324	510

^1^ Risp.: CVI988/Rispens; HVT: Herpesvirus of turkeys; MDV: Marek’s disease virus.

**Table 4 vetsci-11-00139-t004:** Differentially expressed miRNAs in bursae of Fabricius of line 6_3_ and 7_2_ chickens in response to HVT or CVI988/Rispens vaccination followed by a vv+MDV ^1^ challenge.

Treatment ^2^	Chicken Line	miRNA-ID	Log_2_ Fold Change	*p* Value	FDR ^3^
HVT then MDV	Line 6_3_	gga-mir-19b*	12.76	1.34 × 10^−17^	1.92 × 10^−15^
		gga-mir-205b	6.17	7.14 × 10^−6^	6.86 × 10^−4^
		gga-mir-425-5p	3.04	5.70 × 10^−25^	1.64 × 10^−22^
		novelMiR_530	5.87	6.50 × 10^−5^	4.68 × 10^−3^
		novelMiR_547	12.76	1.34 × 10^−17^	1.92 × 10^−15^
		novelMiR_91	−8.07	3.09 × 10^−5^	2.54 × 10^−3^
		novelMiR_215	−6.96	5.00 × 10^−10^	5.76 × 10^−8^
	Line 7_2_	gga-mir-19b*	12.47	6.73 × 10^−6^	9.19 × 10^−4^
		gga-mir-30a*	10.81	6.43 × 10^−6^	9.19 × 10^−4^
		gga-mir-205b	5.87	2.48 × 10^−4^	2.26 × 10^−2^
		gga-mir-425-5p	12.70	1.50 × 10^−6^	8.19 × 10^−4^
		novelMiR_547	12.47	6.73 × 10^−6^	9.19 × 10^−4^
		novelMiR_692	−11.61	2.25 × 10^−4^	2.26 × 10^−2^
Risp then MDV	Line 6_3_	gga-mir-19b*	10.84	2.28 × 10^−6^	1.70 × 10^−4^
		gga-mir-133a-1	1.49	4.94 × 10^−4^	2.10 × 10^−2^
		gga-mir-133a-2	1.49	4.94 × 10^−4^	2.10 × 10^−2^
		gga-mir-425-5p	3.37	2.81 × 10^−22^	8.36 × 10^−20^
		novelMiR_459	5.86	2.18 × 10^−4^	1.18 × 10^−2^
		novelMiR_488	2.54	1.89 × 10^−6^	1.70 × 10^−4^
		novelMiR_530	5.59	3.70 × 10^−4^	1.84 × 10^−2^
		novelMiR_547	10.84	2.28 × 10^−6^	1.70 × 10^−4^
		novelMiR_621-1	5.33	8.87 × 10^−4^	3.52 × 10^−2^
		novelMiR_660-1	7.78	3.16 × 10^−11^	4.70 × 10^−9^
		novelMiR_660-2	7.78	3.16 × 10^−11^	4.70 × 10^−9^
		novelMiR_663-2	5.72	1.57 × 10^−4^	9.34 × 10^−3^
		novelMiR_692	13.94	3.46 × 10^−10^	4.12 × 10^−8^
		novelMiR_91	−8.00	7.64 × 10^−5^	5.05 × 10^−3^
	Line 7_2_	gga-mir-19b*	12.30	1.51 × 10^−150^	2.17 × 10^−148^
		gga-mir-425-5p	13.64	5.18 × 10^−27^	2.98 × 10^−24^
		novelMiR_268	2.89	1.94 × 10^−14^	1.86 × 10^−12^
		novelMiR_547	12.30	1.51 × 10^−150^	2.17 × 10^−148^
		gga-mir-30c-1	−1.33	3.86 × 10^−9^	3.17 × 10^−7^
		novelMiR_91	−9.49	1.50 × 10^−25^	1.73 × 10^−23^
		novelMiR_215	−6.88	2.29 × 10^−4^	1.36 × 10^−2^
		novelMiR_294	−2.97	6.17 × 10^−4^	2.96 × 10^−2^
		novelMiR_692	−11.98	4.95 × 10^−5^	3.56 × 10^−3^

^1^ vv+MDV: very virulent plus Marek’s disease virus; ^2^ HVT: Herpesvirus of turkeys; Risp: CVI988/Rispens; MDV: Marek’s disease virus; ^3^ FDR: False discovery rate.

**Table 5 vetsci-11-00139-t005:** Differentially expressed miRNAs between CVI988/Rispens and HVT vaccination followed by Marek’s disease virus challenge.

Contrast ^1^	Chicken Line	miRNA-ID	Log_2_ Fold Change	*p* Value	FDR ^2^
Risp/HVT	Line 6_3_	novelMiR_508-1	6.32	1.20 × 10^−4^	2.28 × 10^−2^
		novelMiR_508-2	6.32	1.20 × 10^−4^	2.28 × 10^−2^
		novelMiR_508-3	6.32	1.20 × 10^−4^	2.28 × 10^−2^
	Line 7_2_	gga-mir-30a	−11.10	5.79 × 10^−7^	3.30 × 10^−4^
		gga-mir-205b	−6.15	2.53 × 10^−6^	4.80 × 10^−4^
		novelMiR_215	−7.46	2.46 × 10^−6^	4.80 × 10^−4^

^1^ Risp: CVI988/Rispens; HVT: Herpesvirus of turkeys; ^2^ FDR: False discovery rate.

**Table 6 vetsci-11-00139-t006:** Differentially expressed miRNAs between line 6_3_ and line 7_2_ chickens challenged with Marek’s disease virus post-vaccination.

Treatment ^1^	miRNA-ID	Log_2_ Fold Change	*p* Value	FDR ^2^
HVT then MDV	novelMiR_18	7.23	4.24 × 10^−8^	1.17 × 10^−5^
	novelMiR_97-2	2.46	1.42 × 10^−4^	1.63 × 10^−2^
	gga-mir-1684a	−5.85	1.48 × 10^−4^	1.63 × 10^−2^
	novelMiR_215	−7.37	6.26 × 10^−6^	1.15 × 10^−3^
	novelMiR_1062	−7.97	2.24 × 10^−13^	1.23 × 10^−10^
Risp. then MDV	gga-mir-30a	13.51	2.64 × 10^−126^	1.60 × 10^−123^
	novelMiR_203-1	1.16	1.02 × 10^−3^	3.52 × 10^−2^
	novelMiR_203-2	1.15	1.05 × 10^−3^	3.52 × 10^−2^
	novelMiR_508-1	6.43	4.38 × 10^−5^	2.65 × 10^−3^
	novelMiR_508-2	6.43	4.38 × 10^−5^	2.65 × 10^−3^
	novelMiR_508-3	6.43	4.38 × 10^−5^	2.65 × 10^−3^
	novelMiR_660-1	7.90	4.02 × 10^−13^	6.08 × 10^−11^
	novelMiR_660-2	7.90	4.02 × 10^−13^	6.08 × 10^−11^
	novelMiR_663-2	5.84	1.85 × 10^−4^	7.00 × 10^−3^
	novelMiR_692	14.06	6.02 × 10^−12^	7.29 × 10^−10^
	gga-mir-9-1	−1.28	9.47 × 10^−5^	3.84 × 10^−3^
	gga-mir-9-1*	−1.28	9.47 × 10^−5^	3.84 × 10^−3^
	gga-mir-9-2	−1.28	9.50 × 10^−5^	3.84 × 10^−3^
	gga-mir-19b	−1.50	1.39 × 10^−3^	4.01 × 10^−2^
	gga-mir-499	−1.11	1.64 × 10^−3^	4.51 × 10^−2^
	gga-mir-1684a	−6.22	3.69 × 10^−5^	2.65 × 10^−3^
	novelMiR_129-1	−1.28	9.50 × 10^−5^	3.84 × 10^−3^
	novelMiR_129-2	−1.28	9.50 × 10^−5^	3.84 × 10^−3^
	novelMiR_547	−1.50	1.39 × 10^−3^	4.01 × 10^−2^
	novelMiR_600	−8.90	3.47 × 10^−5^	2.65 × 10^−3^
	novelMiR_1062	−8.20	1.34 × 10^−16^	4.05 × 10^−14^
	novelMiR_1108	−5.62	1.19 × 10^−3^	3.79 × 10^−2^

^1^ HVT: Herpesvirus of turkeys; Risp: CVI988/Rispens; MDV: Marek’s disease virus; ^2^ FDR: False discovery rate.

## Data Availability

The small RNA sequence datasets used in this study have been deposited to the SRA website under NCBI with the BioProject Numbers of PRJNA543524 (URL https://www.ncbi.nlm.nih.gov/sra/PRJNA543524; accessed 2 January 2024) and PRJNA544273 (URL https://www.ncbi.nlm.nih.gov/sra/PRJNA544273; accessed 6 January 2024).

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
