# Peer review of "Epigenetic Factor MicroRNAs Likely Mediate Vaccine Protection Efficacy against Lymphomas in Response to Tumor Virus Infection in Chickens through Target Gene Involved Signaling Pathways"

_vetsci, 2024, doi:10.3390/vetsci11040139_

Round 1

Reviewer 1 Report

Comments and Suggestions for Authors

Dear Editor,

Thanks for the opportunity to review this manuscript, and I enjoyed reading this manuscript. It is a well-planned study, and all the findings are presented well. It is a bit misleading to state that microRNAs likely mediate vaccine protection efficacy against Lymphomas in response to tumor virus infection in chickens because the authors presented no experimental evidence to support their claim. I am sure the authors will directly test their hypothesis rather than look for circumstantial evidence to support their claim, in their future studies. 

I understand it will be the next phase of this study and one cannot complete all the studies for one publication. 

Thanks,

Author Response

We thank you for your kind words and professional evaluation of the manuscript. We also appreciate your understanding that it takes time and continuous endeavor to undertake a task like this one: how genetics and epigenetics modulate vaccine protective efficacy in poultry.

Again, thanks,

Huanmin Zhagn

Reviewer 2 Report

Comments and Suggestions for Authors

This manuscript highlights the way by which some expressed MicroRNAs mediate the immune protection induced by two common vaccines against the tumorigeneses of MD infection in two genetically divergent inbred lines of chickens. The current study is really interesting; however, the following comments should be replied:-

1-      Authors should mention the number of birds per each group in the methods section.

2-      The authors indicated that they did euthanasia at 26 days post infection, and then they collected bursa, however, in results (line 303), they mentioned a different euthanasia time point (21 days post infection). Please explain the euthanasia time points clearly in the methods.

3-      In line 209, please change Forty μL to 40 μL.

4-      As known, MD targets the CD4+ T cells, so I think it was worthwhile to analyze the MicroRNAs in thymus rather than bursa of Fabricius; or if the authors may have a rationale for selecting the  bursa.

5-      In MD infections, aerosols can facilitate virus entry into the systemic circulation, including immune organs, by the help of resident macrophages in lung. The authors used the intraperitoneal/ abdominal route in their experimental model. Do you think different routes of infection could impact MD replication with subsequent changes in expression of miRNAs?

Author Response

Dear reviewer,

Thank you for your professional review and evaluation of the manuscript. We appreciate both of your encouragement words and the comments. We made revisions as kindly suggested n your third comment. Please allow me to elaborate on the other comments.

As mentioned in the M&M section, the small RNA sequences were conducted using total RNAs from three sampled birds per treatment group. Yes, there were more birds in each of the treatment groups at the time, but those birds were used for other projects conducted simultaneously at the time to minimize the use of number of birds per project and for the more efficient use of the facility purpose. Therefore, the total number of birds per treatment group at the time was irrelevant to this study and the numbers have been or will be reported accordingly in other project reports.

To the question in comment 2: we regret that it appears confusing but there is no discrepancy.  Where the 21 days were stated (on pages 8 and 9), it is specified as POST (MDV) INFECTION. Whereas the 26 day was stated as POST VACCINATION (on page 4). As stated in the M&M section, vaccination was conducted on the day of hatch; the MDV infection was conducted on the fifth day post hatch.

To the comment 4, the review is absolute right that MD is a T cell lymphoma in response to MDV infection in susceptible chickens. Bursa is primarily a B cell lymphoid organ but studies in mammals have shown that B cell is a KEY player of the adaptive immune response. Our studies also showed that all primary lymphoid organs, not just the thymus, respond to MDV infections and vaccinations. Therefore, we believed that the bursa organ may likely play a role in vaccine protection.

To the comment 5, we share the same curiosity but we can't provide the answer one way or the other with certainty. To ensure repeatability and experimental data's consistence, we have consistently used the intraperitoneal/abdominal route in our experimental model, which, we evaluate, is much more consistent and easier to control the dose effectively received per bird in contrast to the aerosol route approach.

Again, thank you for your support.

Huanmin Zhang